# Non-Markovian Methods in Glass Transition

**DOI:** 10.3390/polym12091997

**Published:** 2020-09-02

**Authors:** Constantino Torregrosa Cabanilles, José Molina-Mateo, Roser Sabater i Serra, José María Meseguer-Dueñas, José Luis Gómez Ribelles

**Affiliations:** 1Center for Biomaterials and Tissue Engineering, Universitat Politècnica de València, 46022 València, Spain; jmmateo@fis.upv.es (J.M.-M.); rsabater@die.upv.es (R.S.i.S.); jmmesegu@fis.upv.es (J.M.M.-D.); jlgomez@ter.upv.es (J.L.G.R.); 2CIBER-BBN, Biomedical Research Networking Center in Bioengineering, Biomaterials and Nanomedicine, 46022 València, Spain

**Keywords:** glass transition, potential energy landscape, dynamic heterogeneity, non-Markovian methods

## Abstract

A model for the heterogeneity of local dynamics in polymer and other glass-forming materials is provided here. The fundamental characteristics of the glass transition phenomenology emerge when simulating a condensed matter open cluster that has a strong interaction with its heterogeneous environment. General glass transition features, such as non-exponential structural relaxations, the slowing down of relaxation times with temperature and specific off-equilibrium glassy dynamics can be reproduced by non-Markovian dynamics simulations with the minimum computer resources. Non-Markovian models are shown to be useful tools for obtaining insights into the complex dynamics involved in the glass transition phenomenon, including whether or not there is a need for a growing correlation length or the relationship between the non-exponentiality of structural relaxations and dynamic heterogeneity.

## 1. Introduction

When a liquid is subjected to cooling, a crystallization process can be induced within it, although in some cases its molecules do not become ordered and the liquid continues with disordered but less mobile structure, and turns into a glass. This process is frequently found in polymers and other materials. It is designated as the glass transition and still remains one of the most significant unsolved problems in condensed matter physics. Although there are several theories that partially explain the phenomena related to glass transition, there are still a lot of questions to be answered (see recent review articles [1,2]).

Computer simulations have been used for decades to study liquids and glasses [3,4]. However, the general purpose molecular dynamics or Monte Carlo methods are complex and require a large amount of computer resources. New algorithms such as the swap Monte Carlo and other related protocols [5,6] have significantly reduced the computation time, although they specifically focus on size polydisperse fluids. Computer models can be extremely detailed and specific or, conversely, coarse-grained when it comes to studying general phenomena. However, even in very simplified models, the evolution of a glass towards equilibrium or the calculation of its own equilibrium state becomes difficult and at very low temperatures requires long computational times.

The molecules that make up a glass are trapped among their neighbors in an environment that hampers their mobility. This low mobility is often considered as the consequence of a highly complex potential energy landscape [7,8,9,10] with high potential barriers between local minima. From another point of view, the Adam and Gibbs’ theory [11] considers a cooperative rearranging region composed of the particles that need to be displaced for a configurational rearrangement to take place. If the temperature of the material falls, the region grows in size, therefore hindering mobility and reducing the probability of transition among the distinct configurations.

By definition, the cooperative rearranging region interacts weakly with its surroundings while its size is variable, and it is on a scale or correlation length that grows or diverges when the temperature is lowered. The variation during its structural relaxation has not yet been solved either experimentally nor theoretically [12,13,14,15,16,17].

As the Biroli and Bouchaud interpretation of the Wolynes et al. argument [18], we consider a small cluster of particles and the boundary conditions imposed by the particles outside the region. Every region jumps from one state to another only because of random fluctuations, thus the supercooled liquid is composed of a mosaic of local metastable configurations. In addition, results from molecular dynamics suggest that transitions involve small compact clusters [19].

The cluster is made up of either simple molecules or small parts of macromolecules, depending on the nature of the material. If only a nanoscale region of an amorphous material is considered, its potential energy landscape of configurations and its energy minima or inherent structures [20,21] is considerably less complex and has been called the local energy landscape [22]. The local energy landscape can be obtained from a single-molecule time series in molecular dynamics, single-molecule spectroscopy, or other techniques [23,24,25,26].

As the cluster is a part of the condensed matter system, it cannot be considered as having a weak interaction or evolving independently of its surroundings. It must be considered as an open system that interacts with other similar adjacent regions or with a random boundary field [18]. According to the notion of dynamic facilitation [27], a slow region can become a fast region and thus exhibit increased mobility when it is adjacent to an unjammed environment and vice versa. The prevailing disorder in the liquid and glassy states generates a broad range of configurations and mobilities for each region and its surroundings that lead to what is known as dynamic heterogeneity [28,29].

Even in homogeneous materials, each cluster is not only exposed to different environments but evolves in time and displays an intermittent behavior characterized by clusters of fast and slow-moving particles [30,31] together with mobile and immobile clusters (as shown in simulations of molecular dynamics) [15] that are reminiscent of the liquidlike and solidlike clusters of the Cohen–Grest free-volume theory [32]. Temporal fluctuations are thought to be related to the dynamics at the glass transition [31] with the emergence of rare, long waiting times between rearrangement events. These dynamics are also characterized by concatenated sequences of rearrangements [33] or intermittent collective displacements of a large number of particles known as avalanches [34]. A softness field has recently been proposed, classifying particles as soft if they are likely to rearrange or hard otherwise [35,36]. In computer simulations, the most important structural feature that contributes to softness is the density of the neighboring regions, so that an increased population on the first-neighbor shell suppresses rearrangements. Experimental testing of softness on colloidal systems has shown how different local environments lead to a distribution of free energy activation barriers that produce non-exponential processes [37].

Markov and higher-order Markov processes [38] are widely used to describe physical phenomena, including glass transition by molecular dynamics and Monte Carlo simulations [4] and have recently been extended also to the study of quantum processes in open systems [39,40,41]. A Markov process is a stochastic process in which the probabilities of transition among the system’s distinct possible states depend only on its current state and not on its previous states. Processes that do not fulfill this property are known as non-Markovian or higher-order Markov processes if the dependence on the past is limited to a finite number of previous states. The occurrence of an event for a second-order Markov process thus depends on the system’s last two states.

For example, the evolution of molecular dynamics in single-molecule fluorescence [42] is described by means of a Markov process between the states of the molecules under study with certain transition probabilities. As the molecule is affected by its environment, non-Markovian or memory effects are often found. To say that the system is non-Markovian means that the description of the system as a Markov process is not complete, as in the case of a molecule that is an open system interacting with its incompletely described environment in condensed matter. In this paper, we propose the application of non-Markovian models to represent and simplify the study of the dynamics of a representative nanoscale cluster of disordered condensed matter. Instead of considering regions inside the material to evolve faster or slower depending on their surroundings, which would force to take into account larger regions, where the evolution of the small region depends on its own history. That history is a consequence of both its internal dynamics and the interaction with its environment. The simplest option is to use second-order Markov models, which have already been applied as a stochastic model of different systems, such as a mesoscopic second-order Markov process for modeling diffusive motion [43], polymerization models [44,45], the study of time series [46], and open quantum systems [47,48].

The aim is to show the extent to which a non-Markovian model of a nanoscale condensed matter region is capable of reproducing the time-dependent and thermal behavior of glass-forming materials. The complex behavior of the material around the glass transition temperature ($T_{g}$) can thus be reproduced by this type of conceptual and computationally simple methods, reducing the system under study to a nanoscale region formed by a cluster of a small number of components coupled to its environment. The complexity of the potential energy landscape representation is notably reduced to the local scale and dynamic heterogeneity is taken into account with the existence of slow and fast clusters with two different values of the potential energy barriers.

The most notable phenomena that characterize glass transition (see [1,2]) are used here as a reference to test the validity of a non-Markovian model of a glass-forming material. A summary of the model’s features can be found below, followed by a comparison of its dynamic behavior with the most important glass transition characteristics [1,2]: its dependence on cooling rate, the stretched exponential shape of the structural relaxation, the anomalous behavior of heat capacity, with characteristic hysteresis and overshoot peaks, and the Kovacs asymmetry and memory effects [49]. Finally, some issues are considered with respect to the stretched relaxations arising fundamentally from the interaction of individual regions with their environment, and the unnecessary existence of a growing length scale, beyond the considered nanometer model scale, to reproduce the considered general phenomena of glass transition.

## 2. Methods

We chose the potential energy landscape perspective as the starting point to define the non-Markovian model of a small region of material made up of a cluster of atoms, small molecules, or molecular segments in polymers. The size of this region has to be small enough to present a small number *n* of possible configurations corresponding to the minima of the local potential energy landscape. The physics of the modeled system was then determined by the local energy minima and the transition paths between them, removing the kinetic energy. The elementary processes were the random transitions between the different inherent structures, and the system evolution was composed of discrete hoping processes [20].

The evolution of systems around glass transition in computer simulations can be represented by complex Markov processes [4] in which the transitions among the distinct configurations are given by certain probabilities that depend on the model. Even the simple two-level system models allow useful insights about glass-forming materials, as in [50] with the evaluation of the configurational entropy and entropy production. Our non-Markovian model of an open nanometer-scale disordered condensed matter region takes into account not only its inherent structures through its potential energy landscape, which determines both multiple energy levels and the intervening potential energy barriers, but also its interaction with adjacent regions, which can facilitate or frustrate transitions among the configurations in the region considered, in line with the dynamic facilitation approach [27].

Transitions in dynamic facilitation models depend on the state of the nearest neighbors. By limiting the study to the time-dependent behavior, the proposed model implements the interaction between the region and its neighbors considering that the region can be surrounded by an environment that is either favorable or unfavorable to rearrangements and that the condition changes according to the presence or absence of previous movements. In the latter case, the region is considered to be in an environment that hampers movements and therefore the potential barriers among the distinct states are considered higher than in the case of the mobile region.

Considering first the evolution of a free region as a Markov process, the transitions from its current state *j* to the following state *k* occurs with transition probabilities $w_{jk}$ at each discrete time interval or Markov step (MS) as a thermally activated process with energy barriers given by a parameter *h*. The process transition rate matrix is
(1)$$w_{jk}\left(T,h\right)=\frac{w_{0}}{n}\exp\left[-\frac{\Delta E_{jk}+h}{k_{B}T}\right]$$
where *n* is the number of energy levels of the region and $w_{0}$ is the reference transition rate to any state at a very high temperature. When $E_{j}>E_{k}$, the energy difference is set to $\Delta E_{jk}=0$. Finally, the no-transition rate is given by $w_{jj}=1-\sum_{k\neq j}w_{jk}$. The transition rate matrix given by Equation (Equation 1) could describe the evolution of a free or closed cluster that could be computed analytically. For example, an adjustable model with an exponential distribution of the fluctuating barrier heights successfully simulated the experimental patterns of slow and fast motions in single molecule spectroscopy [51].

However, in condensed matter, adjacent regions of the material are not independent. The configuration of the neighboring regions largely determines the local density or available volume and therefore the mobility of any region. Each region considered is coupled and interacts with the adjacent regions as an open system with its environment.

As a simple hypothesis, in disordered systems, regions with opposite mobility can be found [36] as a result of their interaction with the distinct adjacent regions (see Figure 1). In each time step, the regions with high mobility, whose surroundings do not hamper their movements, are assigned the transition probability given by Equation (Equation 1) as a simplification, considering their behavior to be the same as the closed region (in terms of mobility). However, other regions may have their mobility and rearrangement probabilities reduced due to interactions with their environment. Within the model, this situation corresponds to an increase of the potential energy barriers to a value $h^{\prime }>h$, a growth of the effective free-energy barriers that can also be found in simulations [52].

In our non-Markovian model, a region that changed its state in the previous Markov step is classified as a fast region, corresponding to the same values of the potential barriers (*h*) as the closed region. In addition, a region that did not change its state in the previous step is classified as a slow region and is characterized by higher energy barriers ($h^{\prime }>h$) between its different configurations. The second-order transition probability $w_{ijk}$ to a state *k* depends on the present state *j* and the previous state *i* as
(2)$$w_{ijk}\left(T\right)=\left\{\begin{array}{cc} w_{jk}\left(T,h\right) & \mathrm{if}\;j\neq i\\ w_{jk}\left(T,h^{\prime }\right) & \mathrm{if}\;j=i \end{array}\right..$$

The difference between the values of the height of the potential barriers assigned to the fast and slow regions can be considered as related to the strength of the coupling between the cluster and its environment and to the material’s dynamic heterogeneity.

A homogeneous disordered material is considered as composed of a set of equivalent regions whose state is described by a distribution vector $d_{i}$, whose components represent the relative occupation of each energy level or, equivalently, the probability of finding a region at each level *i*. For simplicity, in our scenario, there are not different local minima with the same energy level. The initial state chosen, for all the studied processes, was an equilibrium state at high temperature. The equilibrium state of the system at any temperature can be calculated directly, even at lower temperatures, as an eigenvector problem (or related methods). Alternatively, it is often convenient to obtain the equilibrium state by running a brief process of isothermal relaxation at high temperatures. The evolution of the Markov process is calculated at each Markov step, so that the state of the system at the next step is given by its distribution
(3)$$d_{k}^{\prime \prime }=\sum_{i}d_{i}\sum_{j}d_{j}^{\prime }w_{ijk}$$
where $w_{ijk}$ is the transition probability tensor of the non-Markovian process and $d_{i}$ and $d_{j}^{\prime }$ represent the previous and present states, respectively. As a non-Markovian model, the transition probability depends on two previous states, but, in our scenario, it only depends on whether (or not) both previous states are different. In the first case (different previous state), there has been a recent transition and therefore the region has high mobility. The non-Markovianity of the dynamics has already been proposed by some of the present authors (see [33,53]) in relation to Monte Carlo simulations of polymers. However, now the mobility of the region is defined by the height of the energy barriers between the states, and we are able to calculate the equilibrium states at any temperature. Its application has been expanded to a wider range of glass transition phenomena, which will be described below. Computer simulations with this model have been limited to the thermal interaction, i.e., with a fixed set of energy levels, although the height of the energy barriers between them switches between two different values *h* and $h^{\prime }$ due to the interaction of individual regions with their environment.

To evaluate the ability of the non-Markovian models to reproduce the general glass transition phenomenology, both the energy of the equilibrium states and the evolution of the system during different thermal treatments were calculated. The following is a detail of the results of the non-Markovian case model implemented in *Mathematica* with a small set of parameters representing clusters with different degrees of coupling to their environment and compared with a closed cluster. The values of the model parameters in Equation Equation 1 were fixed for simplicity at n=10 energy levels with values at regular intervals between 0 and 1 units of energy and $w_{0}=1$, corresponding to transition probabilities 1/n at very high temperatures.

The values of the potential barrier height between every two states were set to h=0.1 energy units for the different mobile open clusters (Equation (Equation 2)) and the closed cluster (Equation (Equation 1)). The potential barriers of the slow open region were set to higher values, $h^{\prime }=0.2$, 0.5, 1 or 10 energy units. The special case of the potential barriers set to $h=h^{\prime }=0.1$ represents a closed cluster whose potential barriers do not vary and whose evolution fulfills the Markov property. Different open clusters are always defined by h=0.1 and the specified fixed value of $h^{\prime }>h$. According to Equation (Equation 2), their evolution is not Markovian and the higher values of $h^{\prime }$ represent systems with lower degrees of mobility due to the stronger interaction or coupling strength with their environment.

## 3. Results and Discussion

### 3.1. Equilibrium States and $T_{g}$ on Cooling

The energy of the equilibrium states at each temperature was calculated for the closed cluster by means of the resolution of a linear system of equations and for the open systems by solving a system of quadratic equations. The equilibrium energy of the closed cluster against temperature was first compared with that of the open cluster with $h^{\prime }=1$ (see Figure 2). Although their equilibrium energy coincided at very high temperatures (a difference lower than 1% for temperatures higher than T=1), in the region of interest, the equilibrium energy of the open cluster was lower than that of the closed. The only apparent feature in the curve associated with the temperature dependence of the closed cluster was the increased energy at higher temperatures, which is reasonable, since the probability of transitioning towards higher energy levels increases with temperature. However, the possibility of higher energy barriers between the energy levels in the slower open regions has a different effect on the high and low-energy clusters. In the latter case, the higher energy barriers are more likely with lower mobility and hinder the transitions towards higher energy levels. The population of the lower energy levels then increases, corresponding to the lower average energy value. This effect is further accentuated at lower temperatures at which glassy behavior appears. The energy then starts to rise with temperature, arising the glass transition phenomenon, since the probability of transitions towards high energy levels increases at high temperatures.

Interestingly, if an initially closed system is in equilibrium at a given temperature and is then allowed to evolve isothermally and interact openly with its environment, its energy declines with time, i.e., with the number of MS, and approaches a new equilibrium state. If the temperature is low enough (say, below that of glass transition), the system would freeze in a glassy state, depending on the time scale. The isothermal evolution of the open cluster towards equilibrium from the equilibrium states of the closed cluster is shown through the isochronal curves in Figure 2.

Both the equilibrium state and its energy $E_{\infty }$ at any temperature, even those lower than $T_{g}$, can be calculated directly with these Markov and non-Markovian models, and there is no need to determine the time evolution of the system towards equilibrium, a computationally costly operation for a glass with other models.

In the proposed non-Markovian model, the fact that the open cluster has lower equilibrium energy than the closed cluster could be interpreted as the effective linking among clusters of disordered condensed matter due to the interaction that reduces mobility. Surprisingly, in this model, a kinetic parameter such as the height of the potential barriers of the slow regions determines the equilibrium states.

The evolution of the open cluster under cooling sweeps at different rates *q*, in temperature units per MS, from high temperature (T=1) until T=0.01 shows the glass transition of the system as well as the cooling rate dependence of the glass transition temperature (Figure 3). The glass transition temperature $T_{g}$ was calculated from the intersection of the liquid and glass lines shown at the inset of Figure 3. As expected, both $T_{g}$ and the energy of the metastable states increase with the cooling rate. At the higher cooling rates, the energy of the cluster cannot evolve in time towards equilibrium as temperature decreases, reaching a steady value due to the growing relaxation times and the faster increment of the probability of being a slow region.

### 3.2. Isothermal Relaxations

The isothermal energy evolution of the closed and different open clusters was calculated after quenching from a very high temperature (T=3) to the isotherm temperature. Relaxation dynamics was characterized by the energy variations and it depends on temperature, spreading on to several orders of magnitude in time. Figure 4a shows the open cluster (with $h^{\prime }=1$) energy relaxations at different temperatures, with similar curves for the closed and other open clusters. A quasi-exponential time-dependence was found at the higher and the lower temperatures, with the expected slowing down of relaxations as the temperature drops.

In the case of the open clusters, around the glass transition temperature, the curves are non-exponential with a two-step relaxation at the double-logarithmic representation (see Figure 4b). At higher temperatures, isothermal relaxations involve meanly the fast cluster, explaining the single relaxation. As temperature is reduced, the existence of slow and fast regions produces a first relaxation process due to the fast region dynamics at shorter times (before t=10 MS) and a second relaxation at longer times as a consequence of the slow region dynamics. These relaxations can be interpreted respectively as the β and α relaxations of amorphous materials, the latter corresponding to the glass transition, with a typical non-exponential behavior.

Non-exponentiality of the relaxations was characterized by a γ parameter by fitting just the latter part, after t=10 MS, of the energy relaxation results to the Kohlrausch–Williams–Watts function or stretched exponential $A\exp\left[-{\left(t/\tau \right)}^{\gamma }\right]$. The obtained values of the non-exponentiality coefficient γ are shown against temperature in Figure 5 for different systems: the closed cluster and open clusters with different values of the slow region barriers $h^{\prime }$. Values of γ are generally below unity, indicating a distribution of relaxation times for every system due to the transitions among distinct energy levels related to different transition rates, even for the closed cluster. The closed cluster’s non-exponential coefficient is characterized by γ values below 0.7 at the lower temperatures.

Non-exponentiality is different and depends on the value of $h^{\prime }$ for open clusters with different coupling strengths, with lower values of the non-exponential coefficient and a minimum at $T_{g}$ or lower temperatures. In our model, the interaction of the open region with its environment, through the increased value of the potential barriers of the slow regions, is the cause of a pronounced broadening of the relaxation times distribution, with values of the non-exponential coefficient γ lower than 0.5 around $T_{g}$. The lower values of γ are consistent with the higher values ($h^{\prime }$) of the potential energy barriers of the slow regions, which in polymeric materials correspond to chemical bonds between regions leading to a stronger coupling. For other glass forming materials with smaller molecules, the intermolecular interactions would correspond to lower coupling strengths between adjacent regions, with lower values of energy barriers and higher values of the stretching parameter γ.

Dynamic heterogeneity was implemented in this model through the existence of slow and fast regions which can change randomly with two different potential barrier heights. At very low temperatures, most of the regions are slow and their potential barriers are high. At very high temperature, the opposite is true. Between these extremes, there is a temperature interval with a transition from a mostly homogeneous system of slow regions with a high value of the potential barriers’ height, through a dynamically heterogeneous situation with different proportions of slow and fast regions that evolve towards another quite homogeneous system of fast regions at higher temperatures. The obtained results agree with this interpretation, although it would be interesting to calculate some measure of dynamic fluctuations and distributions of relaxation times.

### 3.3. Kovacs Asymmetry and Memory

To check that the open cluster verified the Kovacs’ [49] asymmetry and memory effects, the energy evolution was calculated after diverse thermal treatments. All the processes were initiated from an equilibrium state at a high temperature. To study Kovacs’ asymmetry, temperature single-jumps were performed (0.01 units of temperature with positive and negative sign) from equilibrium states towards the measuring temperature around $T_{g}$, at which the evolution of the energy over time is shown in Figure 6, where the results of the open cluster with $h^{\prime }=1$ at three different temperatures near $T_{g}$ are depicted.

To test the Kovacs’ memory effect, the isothermal evolution of the energy was calculated at T=0.18 after different temperature double-jumps, starting from the system in equilibrium at T=0.19 and with different thermal treatments consisting of isothermal annealing periods for 200 MS at various temperatures under $T_{g}$. It can be seen (Figure 7) that, after double-jump thermal treatments at the lower annealing temperatures, the system initially does not evolve towards the equilibrium energy during the isotherms, in accordance with the Kovacs’ memory effect. After 20 MS, the isothermal energy evolution returns towards equilibrium.

Taking into account the dynamic heterogeneity in a small open cluster model through its elemental fast and slow states, the characteristic Kovacs’ asymmetry and memory effects found in glass-forming materials can be reproduced. These results indicate again that, in spite of the simplicity of the model, a complex underlying distribution of relaxation times is found to adequately describe Kovacs’ effects.

### 3.4. Heat Capacity

Energy evolution of the open cluster with $h^{\prime }=1$ was calculated during heating sweeps at a heating rate of q=0.01 T units/MS after different thermal treatments (see Figure 8a). The previous thermal treatments consisted of cooling from equilibrium at various cooling rates and a range of isothermal annealing periods. On heating after annealing, the glassy state extends to higher temperatures until its energy suddenly approaches equilibrium.

Figure 8b shows the heat capacity, calculated as the derivative of energy with respect to temperature. As normally found in glass-forming materials with differential scanning calorimetry, a heat capacity step is found between the liquid and glass states. Overshoots of the heat capacity produced after annealing are also shown for different annealing times in Figure 8b. The heat capacity overshoots after cooling were also found to depend on the different cooling rates between q=−0.01 and −0.0005 T units/MS.

## 4. Conclusions

This paper proposes a simple stochastic model for a small open region or a cluster of disordered condensed matter designed to reproduce the main phenomena found at glass transition. The interaction of an open region with its adjacent regions was considered to control the high or low mobility of a given region modeled by a non-Markovian process. This model is not based on a given distribution of relaxation times, but the distribution is a consequence of only two different values for the potential energy barriers, the lower for fast regions, and the higher for the slow ones. The results are not equivalent to the simple linear superposition of fast and slow regions. From the dynamic heterogeneity implemented through the second-order Markov process, most of the characteristic behavior of glass-forming materials emerges.

The system’s response to thermal interactions shows a wide distribution of relaxation times. The dynamic heterogeneity seems to be enough to reproduce some of the recognized characteristic phenomena of glass-forming materials: a glass transition temperature that depends on the cooling rate, non-exponential relaxations, the slowing down of the relaxation times with temperature, the Kovacs’ asymmetry and memory effects, and the heat capacity step between the liquid and glass states and overshoots after different annealing conditions.

Therefore, all of these representative features related to the glass transition phenomena can be reproduced by non-Markovian models for a nanometer scale region coupled to its nearest environment, without considering (a priori) a growing correlation length. Nevertheless, when the ratio of slow clusters in the system is high enough, growing aggregates of slow clusters will appear as a consequence, and will even reach percolation at low enough temperatures forming a glass.

Dynamic heterogeneity, implemented in this model only by the potential barrier difference between the fast and slow regions, stretches the isothermal relaxations around the glass transition temperature and produces a very wide distribution of relaxation times. The increased proportion of slow regions as decreasing temperature leads to higher average values of the potential barriers and a pronounced growth of the relaxation times. Thus, the potential barriers of the slow regions define the glass behavior at low temperature, the barriers of the fast regions are related to the liquid behavior at high temperature, and the transition between these two regimes leads to the glass transition phenomena shown here. The complexity of the model’s results around the glass transition suggests the existence of a non trivial underlying distribution of relaxation times that will require further elucidation in future studies.

This minimal model can be useful as a base for more specific microscopic models of materials at glass transition and their structural relaxation processes through the knowledge of their local energy landscape: the local energy levels and barriers between them. It can also be used to examine general questions related to the glass transition from a different point of view as, for instance, the distribution of relaxation times for any system state and its evolution could be directly evaluated in detail from the transition probabilities. The proposed model should be explored both analytically and by computer simulations using a wider set of parameter values than those applied in this initial evaluation, including the effects of pressure [54,55] as a variable controlling the potential energy barriers and even perhaps an attempt could be made to fit it to the data of polymers and other glass-forming materials.

## Figures and Tables

**Figure 1 polymers-12-01997-f001:**
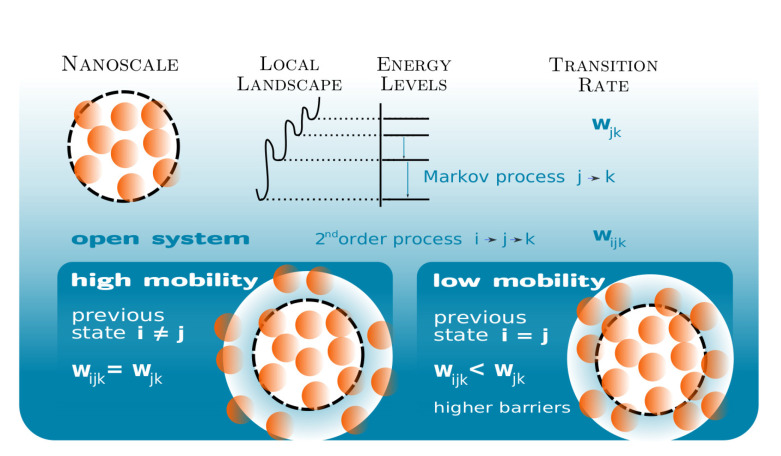
Conceptual diagram of the non-Markovian model of the dynamics of a glass-forming material composed by a set of equivalent clusters. The local energy landscape of the cluster controls its dynamics through the energy levels and the potential barriers between them. The state of the system is given by the fraction $d_{j}$ of clusters at every energy level j. The evolution of the independent or closed cluster (upper panel) is determined by a Markov process with transition probability $w_{\mathrm{jk}}$. The evolution of the open system (lower panel) is more complex: due to the interaction with its environment, the open cluster can have the high mobility of the independent cluster (lower panel, left) or a lower mobility (lower panel, right), maintaining the same energy levels but with higher potential barriers between them. A cluster whose state has changed recently is considered to have high mobility and is considered to have low mobility if it has maintained the same state. The cluster evolution is then a non-Markovian process.

**Figure 2 polymers-12-01997-f002:**
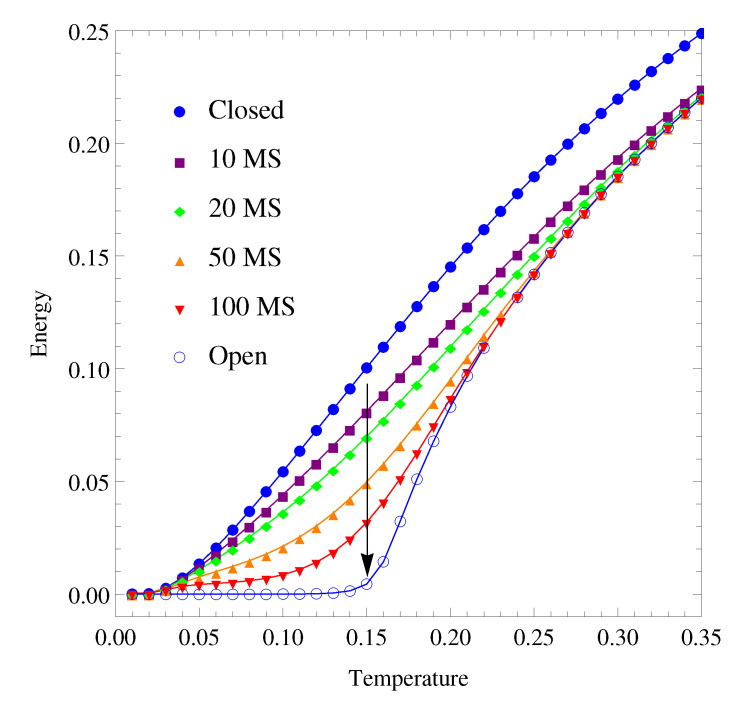
Equilibrium energy per cluster according to the temperature of the closed region with $h=h^{\prime }=0.1$ and of the open system with h=0.1 and $h^{\prime }=1$. Between them, isochronal curves of the isothermal evolutions of an initially closed cluster in equilibrium when opened towards its new equilibrium states (arrow) at different times in Markov steps (MS).

**Figure 3 polymers-12-01997-f003:**
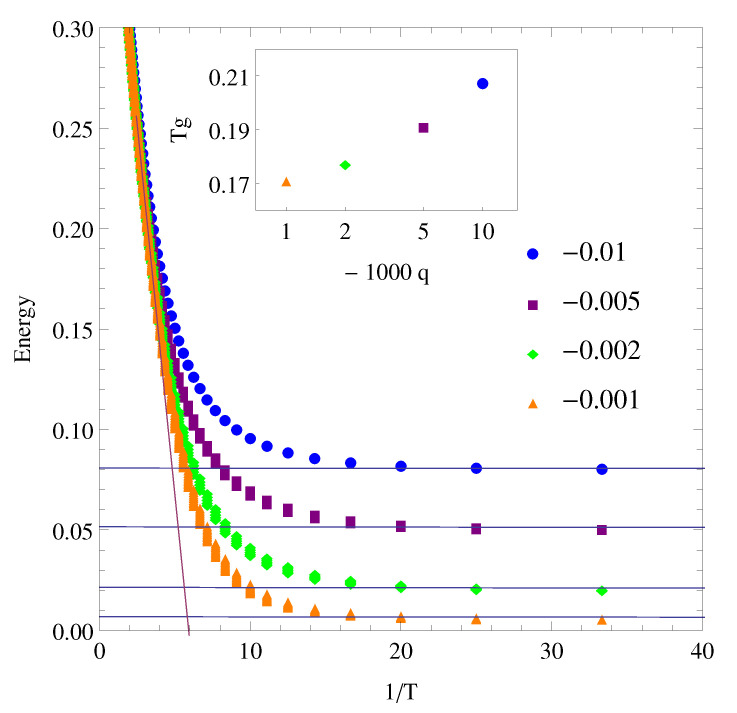
Energy evolution on cooling of an open cluster (h=0.1 and $h^{\prime }=1$) according to the reciprocal of temperature for different cooling rates from q=−0.01 T units/MS to q=−0.001 T units/MS. The liquid and glass lines are included to obtain the $T_{g}$ at the intersection of both lines. The inset shows a glass transition temperature dependence with cooling rate.

**Figure 4 polymers-12-01997-f004:**
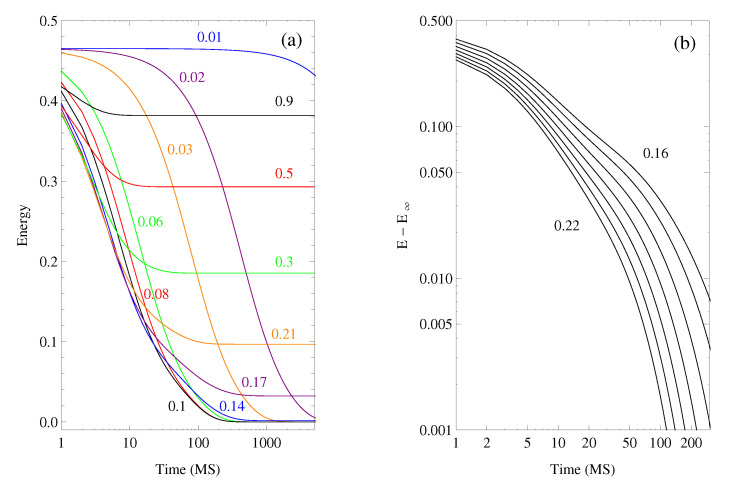
Energy relaxation of the open region. (**a**) evolution at the whole range of temperatures; (**b**) evolution of energy towards its equilibrium value ($E-E_{\infty }$) at temperatures between T=0.16 and T=0.22, every 0.01 T units, shown with more detail at the double-logarithmic representation.

**Figure 5 polymers-12-01997-f005:**
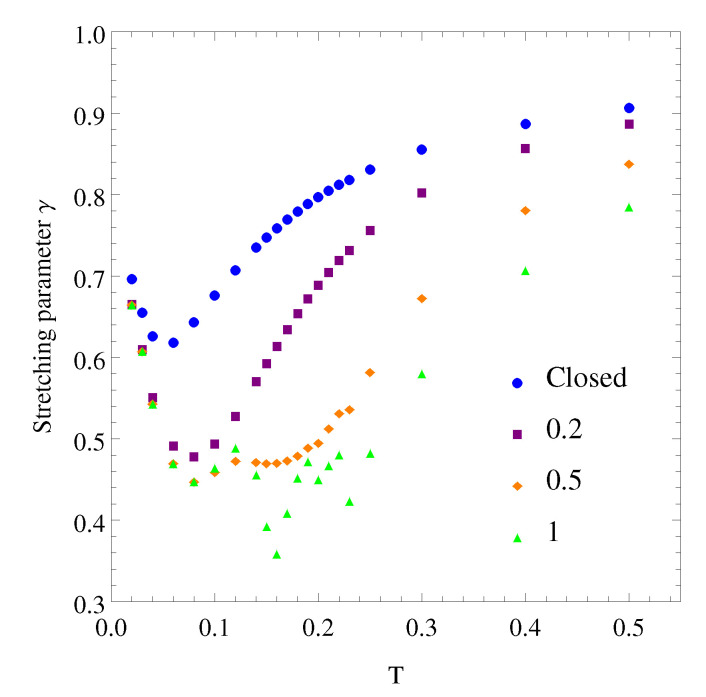
Variation with temperature of the stretching parameter γ obtained from fitting the isothermal energy relaxations for the closed system ($h^{\prime }=h=0.1$) and the open system with different values of $h^{\prime }=0.2,0.5$, and 1.

**Figure 6 polymers-12-01997-f006:**
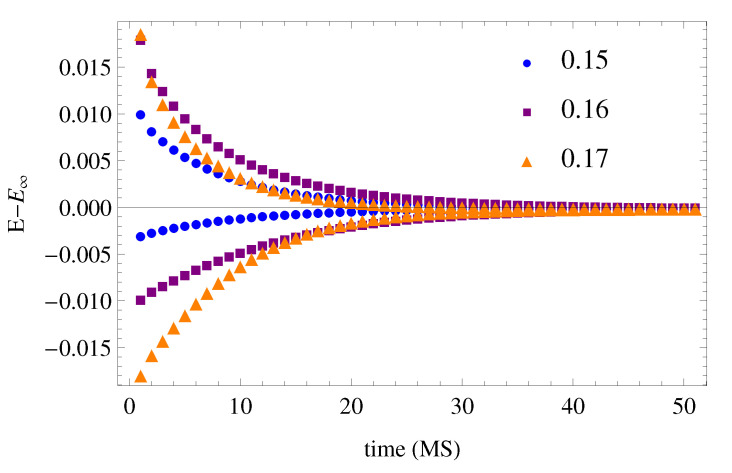
Asymmetry of approach of the of the open cluster ($h^{\prime }=1$) energy E towards its equilibrium values $E_{\infty }$ after temperature single-jumps to T=0.15, 0.16 and 0.17 from temperatures 0.01 units of *T* below and above them.

**Figure 7 polymers-12-01997-f007:**
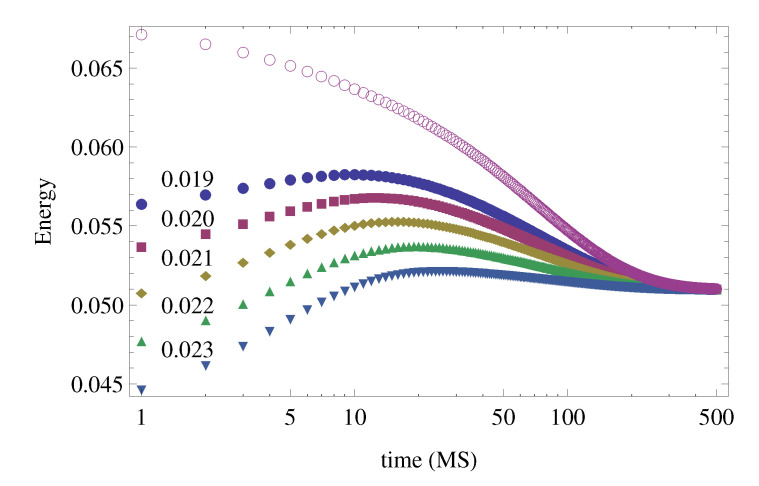
Memory effect at the evolution of the open cluster ($h^{\prime }=1$) energy at T=0.18 after the jump from the equilibrium state at T=0.19 (open circles) and after double-jumps from T=0.19 with an isothermal period for 200 MS at temperatures between T=0.019 and T=0.023.

**Figure 8 polymers-12-01997-f008:**
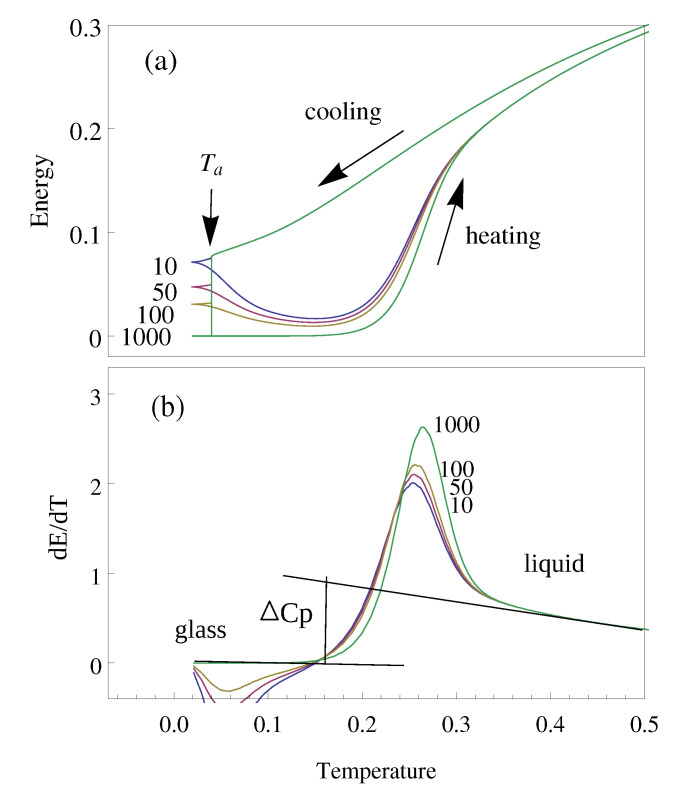
Energy and heat capacity evolution during thermal treatments. (**a**) energy evolution during cooling from equilibrium at q=−0.01 T units/MS, annealing at $T_{a}=0.04$ during 10 to 1000 MS and heating at q=0.01 T units/MS; (**b**) the temperature derivative of the energy, the heat capacity, on heating after the same isothermal annealing periods. In addition, the heat capacity step $\Delta C_{p}$ found between the liquid and glass states is shown.

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
