# Peer review of "Non-Markovian Methods in Glass Transition"

_polymers, 2020, doi:10.3390/polym12091997_

Round 1
Reviewer 1 Report
The paper by C. Cabanilles et al. presents a new model which describes the complex behavor of glass-forming materials upon temperature variation as a minimalistic non-Markovian process of clusters interacting with one another within a heterogeneous environment of neighboring clusters. Somewhat surprisingly, this rather simple stochastic model appears capable of reproducing correctly a large number of characteristic phenomena observed during the glass transition. Most notably, one recovers the familiar dependence of the temperature of vitrification, Tg, and the heat capacity jump on cooling rate, the non-exponential relaxation, some typical memory effects, etc. All these effects are reproduced just by means of only two activation energy barriers between "slow" and "fast" clusters that govern the dynamic heterogeneity of the system.
The manuscript is sufficiently concise, results are adequately supported by figures and the conclusions drawn on the basis of derived results appear plausible and consistent. The current state of the art in the filed is properly reflected by the citations. The paper will certainly be of interest for the vast community of researchers in the field of glassy materials, providing both new insight and incentives for further studies. I recommend, therefore, the publication of this work in the journal of Polymers with two minor revisions:
- An interesting question which is not quite well elucidated in the manuscript concerns the variation of the so called activation volume ΔVa of glass-forming materials with temperature T, which can be determined experimentally from the relaxation time at pressure P, namely, τ(P) = τ0 exp(P.ΔVa / kBT), see C. Roland et al., Rep. Prog. Phys. 68, 1405 (2005). In their Conclusions, on page 13, line 346-348 have mentioned very briefly 'the increased proportion of slow regions' with decreasing temperature but it remains unclear whether this can be viewed as a measure of ΔVa.
- The model is based exclusively on the role of temperature as a governing parameter of the glass transition whereby the impact of pressure remains beyond the scope of this investigation. Does the present model, in addition, permit incorporation of pressure as another independent parameter alongside with T, say, by making the barriers h pressure-dependent, h=h(P), [see W. Xu et al. Macromolecules 2020, DOI 10.1021/acs.macromol.0c01268]?
- Despite Fig.1, the description of the model in terms of the distribution vector di, Eq.(3), is rather abstract and makes it difficult to associate any spatial configuration of the two types of clusters. Are both slow and fast regions of the same size during temperature variation?
Reviewer 2 Report
This article presents enough results and is well organized, but I think the authors should compare their model with other models such as Journal of Non-Crystalline Solids, Volume 533, April 1, 2020, 119907.
Reviewer 3 Report
The authors present a minimal stochastic model for a small open region of disordered condensed matter that is capable to reproduce main phenomena occuring at a glass transition, such as cooling rate dependent glass transition temperature, temperature dependent relaxation times,
the heat capacity step at the transition, Kovacs' asymmetry and memory effects.
All these features of this non-markovian model are achieved considering region on the nanometer scale which is coupled to its nearest environment only through different interactions for regions having high or low mobility. The dynamic heterogeneity of the system is implemented through the 2nd order Markov process.
The model is clearly described, the results are clearly presented, the manuscript is well written to a broad scientific audience, and presumably contains enough new knowledge.
Therefore, I suggest the publication of the paper in Polymers journal, after a revision is considered for few minor changes/corrections as listed below.
1) In Section 3.1. (1st paragraph) the authors mention that the equilibrium energy for the open and closed cluster coincide at very high temperatures (not shown in Fig. 2). At which temperature (T>??) do they coincide?
This question can be important later in Section 3.2., where the system is quenched from the temperature T=3.
Is that temperature above the temperature where the equilibrium energies of open and closed cluster coincide?
2) Looking into the details of Figure 2, it seems to me that at high temperatures (T>0.30) the equilibrium energy for open system with 50MS is slightly lower than those for 100MS and for open system. Is that so,
or it is only an optical illusion? Also in this Figure 2, somehow the word "atoms" appears which should be deleted.
3) In the first sentence of Section 3.2. the authors state that the isothermal evolution was calculated for both the closed and open clusters. However, Figure 4. presents results for the open clusters only, and not for the closed ones. Is there any reason for that?
4) Section 3.2. discusses the isothermal relaxation after quenching. What does exactly the quenching mean in terms of "T units/MS"? If the "quenching rate" is larger than -0.01T units/MS I would expect that the
glass transition temperature is larger than 0.22 (see inset of Figure 3).
5) In Figure 4(a) temperatures are not indicated for all presented curves.
I suggest colour coding of the curves with all temperatures indicated.
